# An inducible CRISPR interference library for genetic interrogation of *Saccharomyces cerevisiae* biology

Amir Momen-Roknabadi [1,2,3,4], Panos Oikonomou [1,2,3,4], Maxwell Zegans [2] & Saeed Tavazoie [1,2,3 ✉]

Genome-scale CRISPR interference (CRISPRi) is widely utilized to study cellular processes in a variety of organisms. Despite the dominance of *Saccharomyces cerevisiae* as a model eukaryote, an inducible genome-wide CRISPRi library in yeast has not yet been presented. Here, we present a genome-wide, inducible CRISPRi library, based on spacer design rules optimized for *S. cerevisiae*. We have validated this library for genome-wide interrogation of gene function across a variety of applications, including accurate discovery of haploinsufficient genes and identification of enzymatic and regulatory genes involved in adenine and arginine biosynthesis. The comprehensive nature of the library also revealed refined spacer design parameters for transcriptional repression, including location, nucleosome occupancy and nucleotide features. CRISPRi screens using this library can identify genes and pathways with high precision and a low false discovery rate across a variety of experimental conditions, enabling rapid and reliable assessment of genetic function and interactions in *S. cerevisiae*.

[1] Department of Biological Sciences, Columbia University, New York City, NY, USA. [2] Department of Systems Biology, Columbia University, New York City, NY, USA. [3] Department of Biochemistry and Molecular Biophysics, Columbia University, New York City, NY, USA. [4] These authors contributed equally: Amir Momen-Roknabadi, Panos Oikonomou. ✉email: st2744@columbia.edu

Technologies that generate systematic genetic perturbations have revolutionized our ability to rapidly determine the genetic basis of diverse cellular phenotypes and behaviors[1–7]. CRISPR-Cas9 technology[8–10], owing to its high fidelity and relatively low off-target effect, has become the dominant method for systematic, high-throughput genetic screening in diverse eukaryotic systems[11–14]. Nuclease-deactivated Cas9 (dCas9) has facilitated genome-wide screens further by enabling transient modulation of target gene expression[15,16]. This is achieved by fusing dCas9 to an inhibitory or activating domain to repress (CRISPRi) or activate (CRISPRa) gene expression[17–22]. At the same time, the use of array-based oligonucleotide synthesis has enabled production of large-scale spacer libraries for use in genome-wide applications. Compared with the traditional genome engineering techniques, such as knock out collections[3,7,23–25], CRISPRi enables the systematic interrogation of all biological processes under different genetic backgrounds and environmental conditions. This technology has been applied in a wide range of organisms from bacteria to human cell lines, to downregulate the expression of both essential and nonessential genes. This has enabled a diverse set of studies from characterizing the role of long non-coding RNAs, to identifying the contributing factors to drug resistance, and many other biological phenomena[16,17,26–29].

One complicating factor in the use of CRISPRi technology is balancing the efficacy of targeting with limiting the off-target activity of CRISPR/dCas9 machinery. Hence, many studies have aimed to determine the rules for efficient gRNA design[16–19]. For example, in K562 human myeloid leukemia cells, optimal gRNAs are found to target a window of −50 bp to +300 bp relative to the transcription start site (TSS) of a gene[26]. In *Saccharomyces cerevisiae*, however, the ideal guide positioning differs from human cell lines. Smith et al.[22] found that the optimal window is a 200 bp region immediately upstream of the TSS. The same group in a subsequent study refined their earlier findings which showed that the region between TSS and 125 bp upstream of TSS is more effective for CRISPR-mediated repression[30]. In addition, they showed a positive correlation between guide efficiency and chromatin accessibility scores.[22] The positioning and design rules of gRNAs are therefore organism specific. Recently, Lian et al.[31] developed a multi-functional genome-wide CRISPR (MAGIC) system for high-throughput screening in *S. cerevisiae*. Although they successfully used a combinatorial approach to map the furfural resistance genes, their system did not utilize yeast-specific design rules[22]. More importantly, their CRISPR system is not inducible. Constitutively expressed library design hinders context-dependent repression of gene function, and makes it difficult to survey the role of dosage-sensitive genes in arbitrary phenotypes of interest. Without inducibility, the cells with spacers targeting dosage-sensitive genes will have a lower fitness as early as the cells are initially transformed. Therefore, an inducible design will help maintain cell populations harboring gRNAs targeting dosage-sensitive and dosage-insensitive genes at the same level. More importantly, it avoids the accumulation of suppressor mutations that could arise during long-term propagation of strains with spacers affecting fitness.

Although Smith et al.[22] introduced a limited diversity inducible CRISPRi system, a genome-wide inducible CRISPRi library is lacking. Here, we introduce an inducible genome-scale library, dedicated for CRISPRi in *S. cerevisiae*, and designed based on the previously described rules[22,30]. We demonstrate the efficacy of this library in targeting essential genes and identifying dosage-sensitive ORFs. In addition, the library enabled us to identify genes involved in adenine and arginine biosynthesis using only a single round of selection. Thus, this CRISPRi library and protocol can be used to efficiently and inexpensively perform genome-wide knockdown screens to discover the genetic basis of any selectable phenotype. In addition, the ability to easily perform CRISPRi screens in a desired genetic background of interest can enable rapid profiling of genetic interactions between a desired allele and knockdowns of all the genes in the genome.

## Results

**Design and construction of a whole-genome CRISPRi library.** We developed our CRISPRi library largely based on the design principles of Smith et al.[22]. They created a single-plasmid inducible system expressing a single gRNA and the catalytically inactive *Streptococcus pyogenes* Cas9 (dCas9) fused to the MXI1 transcriptional repressor[17]. The gRNA is under the control of a tetO-modified RPR1 RNA polymerase III promoter regulated by a tetracycline repressor (tetR), which is also expressed by this plasmid[18,32]. Therefore, the expression of the gRNA is induced by the addition of anhydrotetracycline (ATc) to the growth medium. In addition, a NotI site between the tetO and the gRNA scaffold enables the rapid cloning of spacers. TetR and dCas9-Mxi1 are expressed from the GPM1 and TEF1 promoters, respectively. For compatibility with an ongoing project in our group, we have replaced the *URA3* selection marker in PRS416 with *HIS3*. We call this plasmid amPL43 (Fig. 1a). In order to validate the effectiveness of our system, we cloned gRNAs targeting *ERG25*, *ERG11*, and *SEC14* genes in amPL43. This system demonstrates a low background activity[18] (Fig. 1b, Supplementary Data 1). Upon addition of ATc, the target genes are repressed when compared with the samples without ATc (per qRT-PCR, Supplementary Data 1, Fig. 1c). The repression was seen as early as one hour after induction and could reach as much as 10-fold over a period of 24 hours, depending on the target.

After confirming the effectiveness of this mode of CRISPR-inhibition, we constructed a genome-wide CRISPRi library to target all *S. cerevisiae* genes. To this end, we obtained and ranked all possible spacer sequences targeting every open reading frame (ORF) based on its distance to the TSS and its nucleosome score[22,30]. We then selected the top six gRNAs for each ORF. The dCAS9-MXI1-mediated repression could affect the genes on both plus and minus strands[18]. Therefore, some of the selected sequences could be targeting a neighboring gene with a shared intergenic region. For example, a portion of the gRNAs targeting PTA1 could affect ERV46 (Fig. 1d). For genes that share a gRNA with another gene, we selected up to six additional sequences unique to those genes (see Methods). In order to evaluate the guide design parameters, we included ~5000 gRNAs that target further upstream of TSS or downstream of TSS. Altogether, we designed >51,000 gRNAs, with between 6 and 12 sequences per gene (Supplementary Data 2). As a negative control for gRNA activity, we introduced 500 synthetic randomly shuffled gRNAs with no matches in the yeast genome (Supplementary Data 3).

This oligonucleotide library was synthesized on a 92918-format chip and cloned into amPL43 using universal adapter sequences. In brief, the pooled oligonucleotides were amplified by PCR, cloned using Gibson Assembly[33], and transformed into DH5α *E. coli* (>100× colonies/gRNA). The transformed bacteria were grown in a semisolid LB as individual colonies to minimize competition between the strains. Semisolid 3D media provides a more cost-effective and less labor-intensive method for large-scale libraries than conventional 2D plating. The pooled plasmid library was transformed into BY4741 using the standard LiAc/PEG method[34] with minor modifications, and grown in semisolid SC-His+glu media (two or three biological replicates, Supplementary Fig. 1) for 48 h, pooled, and resuspended in SC-His+glu media, and frozen for future use.

The short gRNA sequences can act as unique identifiers of individual strains and, like barcodes, can be quantified using

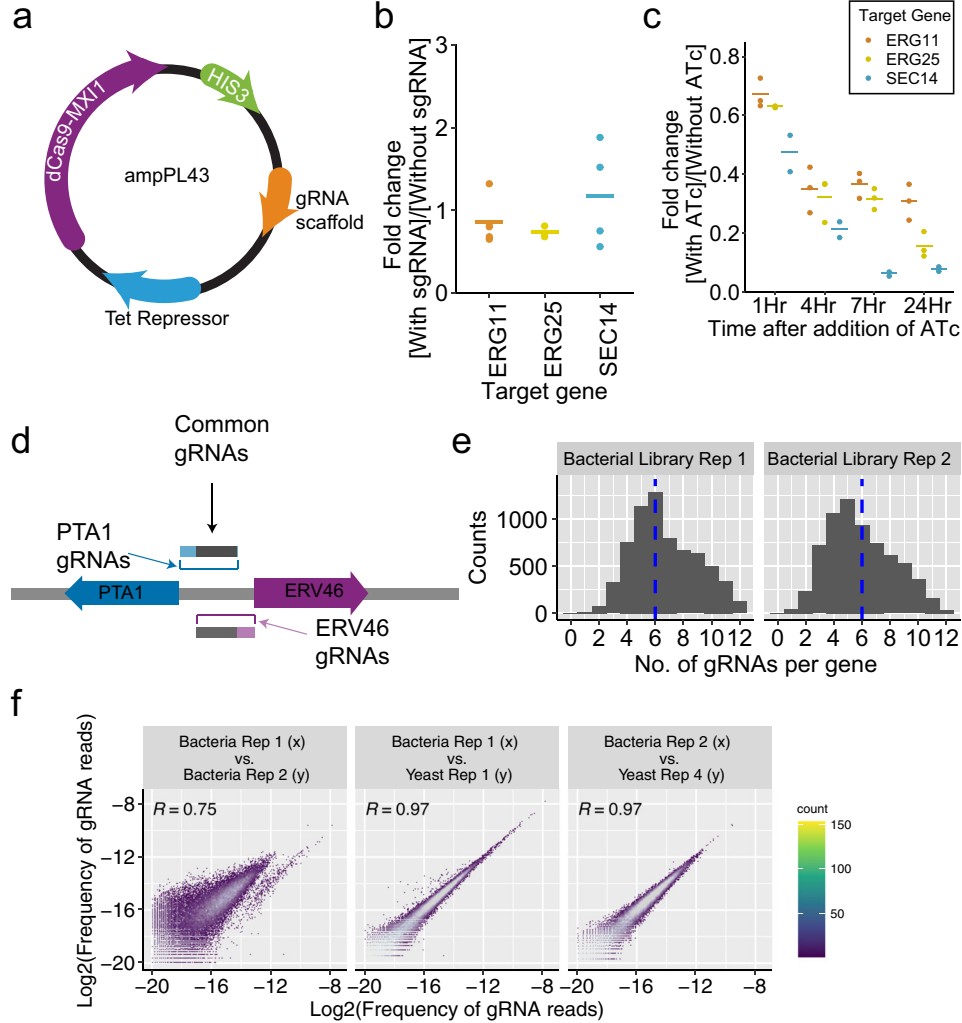

**Fig. 1 CRISPRi library design and properties. a** Schematic of amPL43 expression vector for inducible CRISPRi library in *S. cerevisiae*. **b** The expression fold change of each target: ERG25 (three replicates), ERG11 (five replicates), and Sec14 (four replicates), as a result of presence of sgRNA, without induction after 24 hours, as measured by qPCR. The mean for each sample is represented by a solid line. **c** The expression fold change of each target: ERG25 (three replicates), ERG11 (three replicates), and Sec14 (two replicates), as a result of gRNA induction by ATc, was calculated over time by qPCR. The mean for each sample is represented by a solid line. **d** Schematic depicting the genomic region of PTA1 and ERV46. The gRNAs targeting the region between the two genes, depending on their proximity to each gene, could affect both genes. **e** Histogram depicting the number of gRNAs per gene in two library replicates. The dashed blue line denotes the median: ~6. **f** Scatter plot depicting the frequency of reads per gRNA between select biological replicates of the CRISPRi library. Pearson Correlation *R* value is reported for each pair.

next-generation amplicon sequencing. To prepare the library for sequencing, the plasmids were extracted from yeast samples, and the targeting region was PCR amplified with flanking Illumina adapters and submitted for sequencing. We generated two independently cloned biological replicates of the library in bacteria. At the realized sequencing depth, our first replicate included >41,000 gRNAs, targeting all of the yeast ORFs except YBR291C, while >36,000 gRNAs were present in the second replicate, targeting all but five of the yeast ORFs (YBL023C, YIL171W, YJL219W, YOR324C, YPR019W; Fig. 1e and Supplementary Fig. 2). The two bacterial replicates are highly correlated (Fig. 1f, Pearson correlation $R = 0.75$). The loss in the number of gRNAs after cloning could be attributed to limited sequencing depth (~7 million reads for the first replicate and ~4 million reads for the second replicate) or synthesis errors. Biological replicates from the two bacterial libraries were transformed into yeast. There is a strong correlation between the frequency of reads of the gRNAs in the bacterial plasmid library replicates and those

transferred to yeast demonstrating that there is no systemic bias as a result of the transformation. (Fig. 1f, Supplementary Fig. 3). The five separate biological replicates of the yeast library gave highly reproducible diversity and abundance (Supplementary Fig. 4 and Supplementary Data 4). The gRNA library does not show a bias for any specific GO Term and shows good representation across compartments, functions, and biological processes (Supplementary Fig. 5).

**High-throughput identification of dosage-sensitive genes**. To demonstrate the utility of our CRISPRi library for high-throughput genotype–phenotype mapping, we set out to determine whether we could systematically discover dosage-sensitive genes by using a simple outgrowth experiment. We focused on determining how well these dosage-sensitive genes corresponded to genes previously found to be haploinsufficient in yeast[2]. One of the major advantages of our library is its inducibility. Inducible

gRNA expression allows us to efficiently target dosage-sensitive genes for short intervals and determine their phenotypic outcome. To this end, we inoculated semisolid media with the equivalent of 1 OD660 of the library (with an average of ~450 copies of each library member) with and without ATc induction (250 ng/mL). The use of semisolid media reduces direct competition between strains, helping to maintain a more uniform representation of gRNAs. The fitness consequence of knocking down every yeast gene can be determined using the depletion or enrichment of the barcodes in the library in a comparison of induced and uninduced samples. Under these conditions, we expect that the gRNAs targeting haploinsufficient genes should be significantly depleted in the induced library (ATc+).

We grew three library replicates for 24 h, extracted plasmids from pooled samples, and prepared amplicons for next-generation sequencing (see Methods). We calculated the log-fold change of reads between samples with and without ATc, for these replicates. To calculate the statistical significance of gRNA depletion, we simulated a library of synthetic scrambled genes by sampling the scores from the synthetic randomly shuffled gRNAs to create a baseline (see Methods). Not all gRNAs will efficiently repress the expression of their target gene[22]. Therefore, we only focused on the impact of the most effective gRNAs for a given gene. To this end, we sorted the gRNAs for each gene based on their ratio and designated the mean of the most effective three gRNAs as the score for that gene (see Methods). We used the synthetic scrambled gene distribution to calculate z scores from the gene depletion scores for each replicate. Next, we averaged the z scores for the biological replicates using Stouffer's method. The gene depletion scores between the replicates were well correlated (Fig. 2a). This correlation is much stronger for the haploinsufficient genes, which demonstrates the reproducibility of the repression as the result of gRNA induction (Fig. 2b). The gRNAs targeting known haploinsufficient genes are significantly depleted in the induced sample compared with the uninduced sample ($p$ value $< 1 \times 10^{-15}$, Wilcoxon signed-rank test). We used the background distribution of synthetic scrambled genes to define a depletion score threshold, measure false discovery rate (FDR), and determine which genes are significantly affected as the result of gRNA induction at a given FDR (see Methods). With only a single round of growth selection, and three replicates, we were able to correctly categorize ~85% or ~81% of haploinsufficient genes with FDR < 10% or FDR < 5%, respectively (Fig. 2c, d and Supplementary Data 5 and 6).

A major advantage of our inducible CRISPRi system is the ability to interrogate the role of essential genes in any selectable phenotype. In order to explore this capacity, we set out to determine what percentage of known essential genes are also dosage sensitive under our experimental conditions. The sensitivity and specificity of CRISPRi-based discovery of essential genes are limited by fundamental biological factors. On the one hand, although essential genes are associated with functions that are indispensable to cellular life, it has been shown that not all essential genes are dosage sensitive[2]. This implies that reducing the dosage of even some of the essential genes to 50% would not measurably affect cellular fitness. On the other hand, essential genes are overrepresented among dosage-sensitive genes[2]. Therefore, it is of interest to determine what percentage of known essential genes show dosage sensitivity and therefore can be detected by systematic CRISPRi knockdown. Indeed, we observed that the majority of essential genes exhibit significantly lower gene depletion scores (Fig. 2d, $p$ value $< 1 \times 10^{-15}$, Wilcoxon signed-rank test). Overall, we observed that ~67% of essential genes show dosage sensitivity based on their gene depletion scores (FDR < 10%, Supplementary Fig. 6a). Surprisingly, our analysis showed that 37% of nonessential genes show dosage sensitivity.

However, 76% of the nonessential genes detected here as dosage-sensitive have been previously shown to decrease fitness when mutated[5,35]. Therefore, our results are in line with a low false discovery rate, bolstering the utility of this library for systematic genetic analysis of phenotypes.

Next, we explored the association of factors such as strandedness, distance to TSS, secondary structure free energy, and nucleosome occupancy score to the gRNA depletion score in our library in a systematic fashion. As can be seen in Fig. 3a, gRNAs located between TSS and 150 bp upstream of TSS are particularly effective for detecting dosage-sensitive essential genes, with the strongest effect for the guides targeting 50 bp upstream of TSS, thus further refining yeast-CRISPRi design rules[30]. In addition, as may be expected, a higher nucleosome score seems to reduce the effectiveness of gRNA mediated transcriptional repression, with the most effective gRNAs having a nucleosome score near zero (Fig. 3b). As for the gRNAs targeting nonessential genes, we did not observe any dependency between the gRNA depletion score and the distance to TSS or the nucleosome occupancy score (Fig. 3c, d). In addition, we observed an inverse association between the stability of the gRNA's secondary structure and its efficacy, for both essential and nonessential genes (Supplementary Fig. 6b, c). Finally, we investigated whether gRNAs' effectiveness in the context of our library is influenced by the strandedness of gRNA targeting. To minimize the effect of other factors, we only focused on gRNAs targeting dosage-sensitive essential genes that have a nucleosome occupancy score less than 0.1 and are within 125 bp upstream of TSS. The mean of the gRNA depletion score for gRNAs with the PAM on the same strand as the ORF was $-1.32$ while the mean for the gRNAs on the opposite strand was $-0.95$ (Fig. 3e, $p$ value ~$4.2 \times 10^{-7}$, Wilcoxon signed-rank test). In addition, the gRNAs with the PAM on the same strand as TSS are most effective when located between 50 and 75 bp upstream of the TSS, whereas the gRNAs with the PAM on the opposite strand, have a maximum efficacy between 25 and 50 bp upstream of the TSS. In addition, the most effective gRNAs with the PAM on either strand have lower nucleosome occupancy scores (Fig. 3f–i).

Finally, to systematically assess the factors contributing to gRNA efficacy, we developed a random forest model to classify effective guides based on the distance of PAM to TSS, sequence features (GC Content, longest run of each poly nucleotide, mono- and di-nucleotide composition at each position), nucleosome occupancy score and stability of sgRNA. We trained a classifier to discriminate between the most effective 20% and the least effective 20% of gRNAs targeting dosage-sensitive essential and haploinsufficient genes. To avoid overfitting, we implemented threefold cross-validations and repeated the process for 10 different partitions. Our model is highly predictive of activity in the test set of genes, with an area under the curve (AUC) of $0.874 \pm 0.005$ and area under the precision–recall curve (AUPRC) of $0.862 \pm 0.005$ (Fig. 4a, b, only the best replicate is shown). The class prediction probability showed a Spearman correlation ~0.51 with the observed score (Fig. 4c).

We next analyzed which features contributed most to the predicted activity in the classifier (Fig. 4d, see Methods). Overall, the distance of PAM to the TSS had the greatest percentage contribution (~54–65%, $p$ value $< 10^{-10}$, Wilcoxon signed-rank test), whereas overall sequence features represented the second largest effect on the model (~34–49%, $p$ value $< 10^{-10}$, Wilcoxon signed-rank test). Nucleosome occupancy contributed to a lesser extent (~2–8%, $p$ value $< 10^{-3}$, Wilcoxon signed-rank test), whereas additional individual parameters including gRNA secondary structure, strandedness and GC content were not deemed as significant by the model ($p$ value $> 0.1$, Wilcoxon signed-rank test). We next investigated the contribution of each

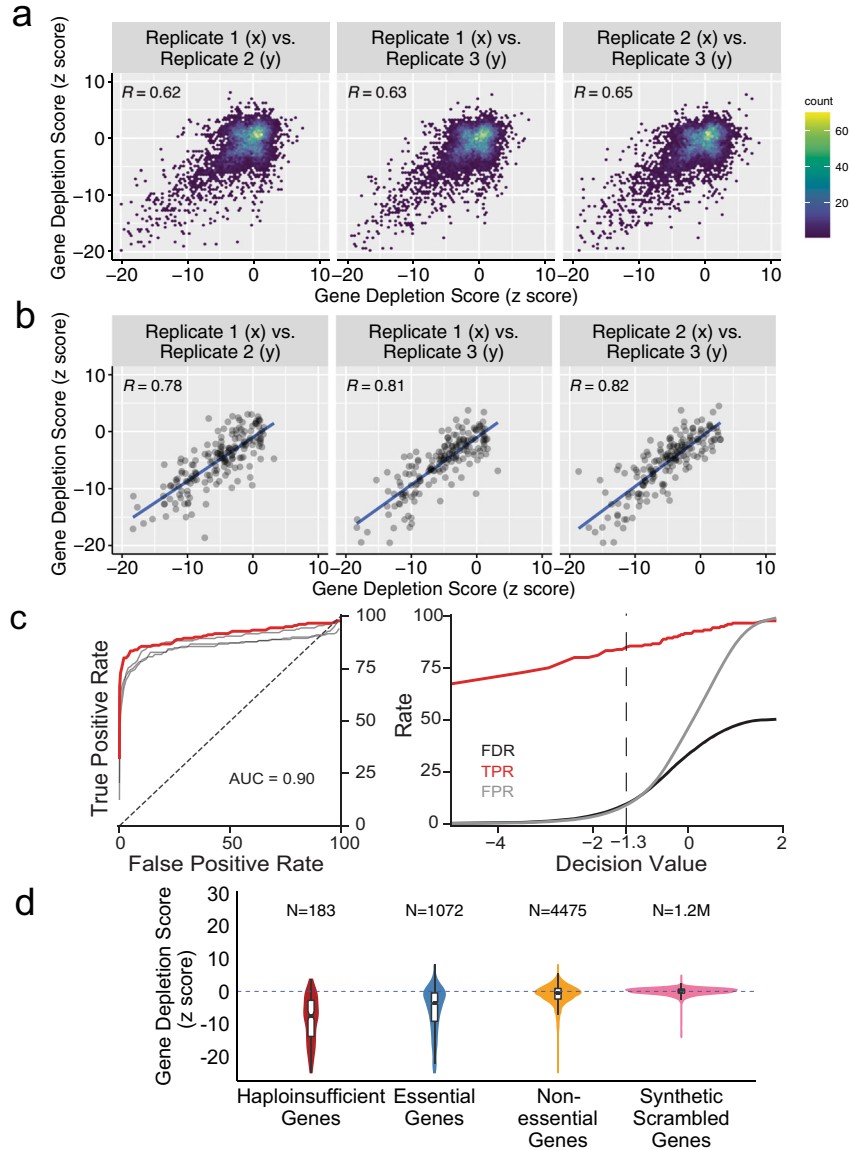

**Fig. 2 High-throughput identification of haploinsufficient and dosage-sensitive genes. a** Binned scatter plots of gene depletion score between the replicates. Pearson Correlation $R$ value is reported for each pair. **b** Binned scatter plots of gene depletion score, limited to haploinsufficient genes, between the replicates. Pearson Correlation $R$ value is reported for each pair. **c** The Receiver Operating Characteristic Curve (ROC) for the detection of haploinsufficient genes and FDR, TPR, and FPR trends based on decision values. ROC curve shows that the depletion $z$ score is a strong classifier for haploinsufficient genes. The individual replicates are shown in gray. Area under the curve is 0.90. Dashed line denotes the decision value for FDR < 10%. **d** Violin plots depicting the gene depletion score distribution for the haploinsufficient genes, essential genes, nonessential genes and synthetic scrambled genes, resulting from 200× random sampling of synthetic randomly shuffled gRNAs, in induced (SC-His+ATc) versus uninduced samples (SC-His-ATc).

nucleotide at each location between 10 bp upstream and downstream of the target spacer (Fig. 4e, f). This analysis demonstrated that the bases located at 1, 2, 4, 9, 10, 11, 12, and 20 base pairs upstream of PAM contribute significantly to the efficacy of gRNA ($p$ value < 0.01, hypergeometric test). Out of these positions, gRNAs with an "A" at position −9 are predicted to have the highest contribution to an effective gRNA. In addition, the gRNAs without a poly-T stretch are more effective.

The classifier presented here detects important biological features in the context of the library. However, this model has some inherent limitations owing to the limited range and interdependence of our library's features. Our library was designed based on the rules previously described by Smith et al.[22], and therefore the features of the library have a limited diversity. For example, the gRNAs are mostly located within 200

bp upstream of the TSS and nucleosome occupancy scores were low. The strong bias toward optimal features could in part explain why the strandedness was not deemed to have a significant predictive power for the model ($p$ value > 0.1, Wilcoxon signed-rank test). Furthermore, features such as GC content, nucleotide compositions at each position, and stability of secondary structure can be highly interdependent. Since the gRNAs in this library were not sampled in an unbiased way to randomly cover all available PAMs in yeast, the results of this predictive model should be interpreted as significant within the context of this specific library. Nevertheless, the model captures some of the biological rules that have been observed previously for other organisms. Compared with the contributing factors for the sgRNAs' predicted efficacy in human cells[36], our model shows that although the distance of PAM to TSS is the most important

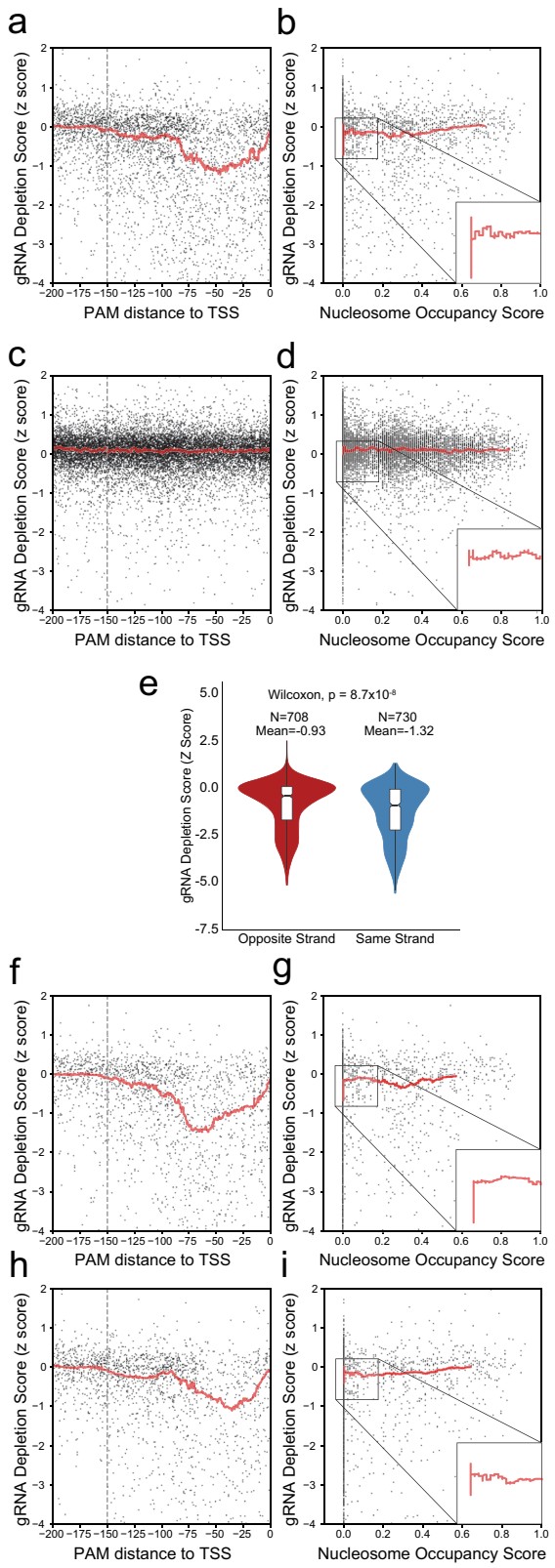

**Fig. 3 Correlation of gRNA depletion score with distance to TSS, nucleosome occupancy score, and strandedness. a** Scatter plot depicting the gRNA depletion score for gRNAs targeting dosage-sensitive essential genes versus the distance from TSS. The red line represents the rolling average (window of 200). The dashed line signifies 150 bp upstream of TSS. **b** Scatter plot depicting the gRNA depletion score for gRNAs targeting dosage-sensitive essential genes versus the nucleosome occupancy score. The red line represents the rolling average (window of 200). **c** Scatter plot depicting the gRNA depletion score for gRNAs targeting nonessential genes versus the distance from TSS. The dashed line signifies 150 bp upstream of TSS. The red line represents the rolling average as above. **d** Scatter plot depicting the gRNA depletion score for gRNAs targeting nonessential genes versus the nucleosome occupancy score. The red line represents the rolling average as above. **e** Violin plots depicting the distribution of gRNA depletion scores for gRNAs on the opposite or same strand as the target ORF. **f** Scatter plot depicting the gRNA depletion score for gRNAs targeting dosage-sensitive essential genes with PAM on the same strand as TSS versus the distance from TSS. The red line represents the rolling average as above. The dashed line marks 150 bp upstream of TSS. **g** Scatter plot depicting the gRNA depletion score for gRNAs targeting dosage-sensitive essential genes with PAM on the same strand as TSS versus the nucleosome occupancy score. The red line represents the rolling average as above. **h** Scatter plot depicting the gRNA depletion score for gRNAs targeting dosage-sensitive essential genes with PAM on the opposite strand as TSS versus the distance from TSS. The red line represents the rolling average as above. The dashed line marks 150 bp upstream of TSS. **i** Scatter plot depicting the gRNA depletion score for gRNAs targeting dosage-sensitive essential genes with PAM on the opposite strand as TSS versus the nucleosome occupancy score. The red line represents the rolling average as above.

overfitting. Therefore, the random forest classifier could be used to design a more effective library in the future.

**Identification of the genes involved in biosynthesis of adenine and arginine.** Next, we explored whether our CRISPRi-based approach can efficiently identify smaller sets of genes associated with specific biological processes. We, thus, chose to identify the genes involved in two distinct biosynthetic pathways, arginine and adenine. This was done by CRISPRi profiling, across five replicates, of induced cells grown in drop-out media for arginine and adenine, compared with cells grown in media including the nutrients. As a necessary condition for plasmid maintenance, histidine is absent in both conditions. We expected gRNAs-targeting genes that contribute to general cellular fitness to be depleted to a similar degree for both samples, whereas gRNAs involved in the biosynthesis of arginine and adenine to be differentially affected. As such, we quantified the depletion of gRNAs in the arginine/adenine/histidine drop-out media against the histidine drop-out media control. In order to systematically determine all the pathways that were affected, we used iPAGE[37], a sensitive pathway analysis tool that directly quantifies the mutual information between pathway membership and the global distribution of gene depletion scores. The iPAGE results show that the biological processes for arginine biosynthesis and purine biosynthetic processes[38–44] are significantly informative of the depletion of gRNAs targeting the pathways ($p$ value < 0.05, random shuffling, Supplementary Fig. 7 and Supplementary Data 5 and 6). In addition, the gRNAs targeting the general pathways for ATP export, nitrogen starvation, and amino-acid biosynthesis were also depleted. The iPAGE analysis also showed that gRNAs targeting pathways affecting protein dynamics, such as translation

factor in the sgRNA activity, the predicted secondary structure and homopolymers, besides Poly-Ts, are not as important. Furthermore, the contribution of each individual parameter was relatively consistent across replicates, suggesting that the model is indeed identifying underlying biological factors and not simply

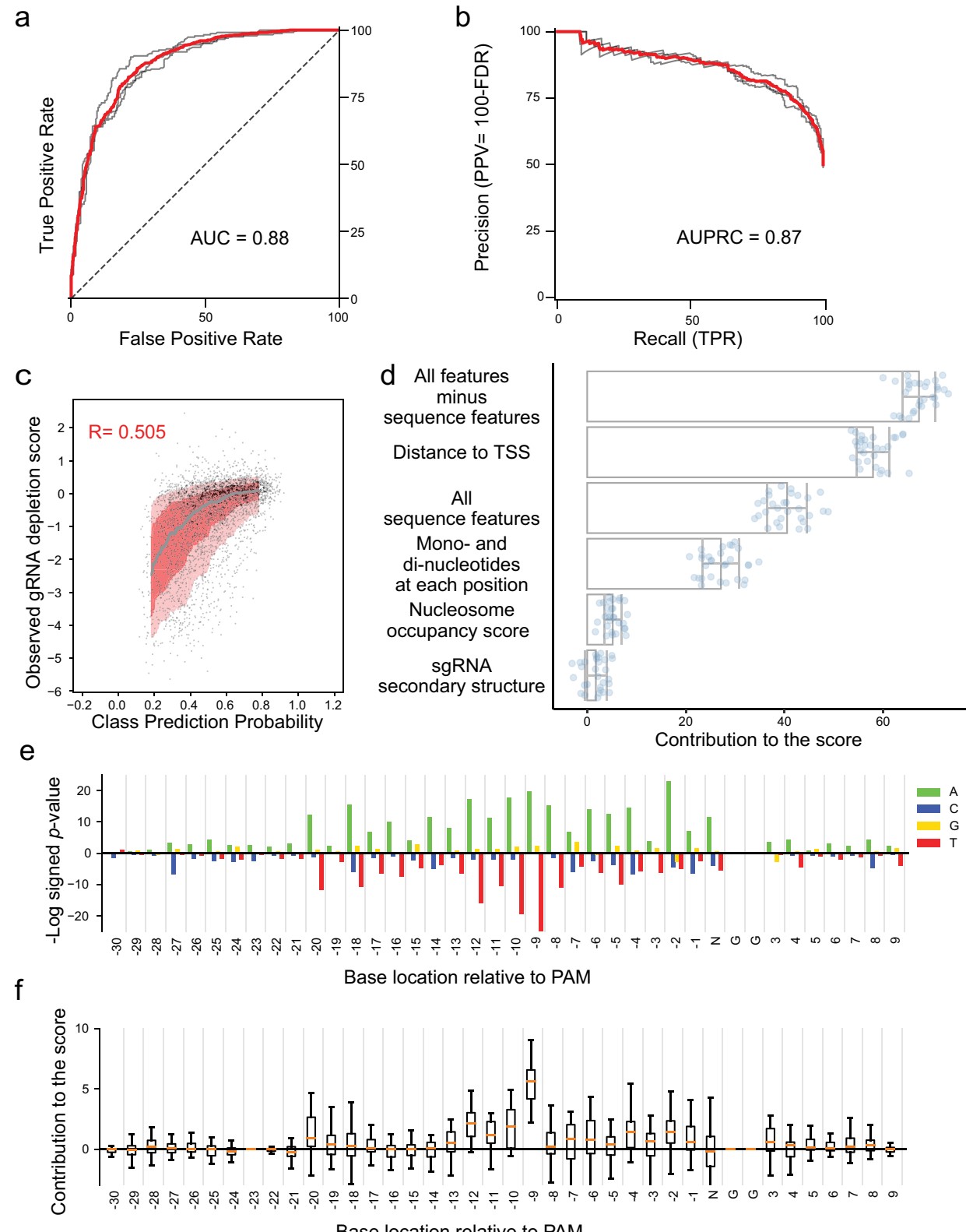

and import of proteins into the nucleus, are significantly enriched ($p$ value $< 0.05$, random shuffling). This suggests that the activity of these pathways partially affect growth fitness when cells are simultaneously deprived of arginine and adenine. We also detected significant depletion/enrichment of pathways associated with protein sorting, such as endosome to Golgi transport and retrograde vesicle transport ($p$ value $< 0.05$, random shuffling).

This is in line with previous observations that mutations in many of the arginine biosynthesis genes are known to cause abnormal vacuole morphology[45]. Our data, thus, provide additional evidence that arginine deficiency undermines protein sorting functions in *S. cerevisiae*.

Our screen detected 11 out of 19 genes annotated in core arginine biosynthesis and adenine biosynthesis pathways (Fig. 5a,

**Fig. 4 A random forest model can classify efficient gRNAs. a** The receiver operating characteristic curve (ROC) for the classification of efficient sgRNAs using random forest classification (one of the ten replicates). ROC curve shows strong classifier performance. The individual trends for the three cross-validation models assessed on their respective testing set are shown in gray. Area under the curve is 0.88. **b** The Precision–Recall curve for the classification of the efficient sgRNAs using random forest (one of the ten replicates). The individual trends for the three cross-validation models assessed on their respective testing set are shown in gray. Area under the curve is 0.87. **c** Class prediction probability vs. observed gRNA depletion score for the random forest classifier. The class prediction probability is based on the probability of the gRNA being classified as efficient by the classifier, and therefore does not have the same range as the observed gRNA depletion scores. The spearman correlation is 0.505. **d** Percentage contribution of the features to the predicted efficacy score of gRNA. The features were the distance of PAM to TSS, sequence features (GC Content, longest run of each poly nucleotide, mono- and di-nucleotide composition at each position), nucleosome score and stability of sgRNA. The error bars are equivalent to the standard deviation. **e** -Log of signed $p$ value of the over and under representation of each nucleotide in every position for the top 20% gRNAs predicted to be efficient compared with all the gRNAs in the model. For each position, we inferred the significance of having A, G, C, or T at that position. $P$ value was calculated using a hypergeometric test. **f** The contribution of each base to the efficacy of gRNA was calculated by shuffling the nucleotides at that position between all of the test set. The box extends from the lower to the upper quartile values, while whiskers extend 1.5 times the interquartile range from the edge of the box.

---

b, Supplementary Fig. 6d, FDR < 10%). Arginine biosynthesis is a particularly complex biosynthetic pathway with connections to several other pathways, such as polyamine and pyrimidine biosynthesis, and certain degradative pathways[40,46]. In addition, transcription of arginine biosynthetic genes is repressed in the presence of arginine by the ArgR/Mcm1p complex, which consists of Arg80p, Arg81p, Arg82p, and Mcm1p[47]. As such, downregulation of the ArgR/Mcm1p complex genes would be expected to increase the fitness of the cell in the absence of arginine in the media. Consistent with this expectation, we found that the gene depletion scores for these genes were positive, providing further support for the sensitivity of our CRISPRi screen to detect positive and negative contributors to a phenotype of interest (Fig. 5a). Our analysis of gRNA efficiency showed that the gRNAs with PAM 0–150 bp upstream of each TSS are particularly effective. Therefore, we explored whether we could improve the precision of our detection by limiting our analysis to gRNAs that target 0–150 bp upstream of the TSS, while maintaining gRNA diversity. As is shown in Fig. 5c, we detected 15 out of 19 adenine and arginine biosynthesis genes (FDR < 10%, Supplementary Data 5).

It is important to point out that our FDR estimate of 10% is conservative, since strict gene ontology membership of these pathways does not capture the full complexity of the highly interactive genetic and regulatory networks coordinating nucleotide and amino-acid metabolism. For example, in addition to the genes involved in arginine and adenine biosynthesis, most depleted genes were involved in protein sorting and other general amino-acid metabolic pathways such as SSY1 which is known to sense external amino-acid levels[48].

## Discussion

Here, we have established a versatile genome-wide functional screening library for CRISPRi in S. cerevisiae and further refined the gRNA design rules for efficient transcriptional repression. The ability to interrogate essential genes, led us to discover that 67% of essential genes also exhibit dosage sensitivity under our experimental conditions.

Genetic studies in *Saccharomyces cerevisiae* benefit from a wide array of techniques for studying loss-of-function phenotypes. Some of the most widely used methods to study loss-of function in budding yeast are gene deletion/knock-outs[3,7,23–25], temperature-sensitive mutants[49,50], and DAmP mutants[35,51]. However, temperature-sensitive mutations are difficult to construct in a systematic manner. Gene deletion libraries can be constructed in a systematic manner, although essential genes will not be covered, at least, in a haploid background. Finally, in DAmP library assays, only 17% of the viable alleles for essential genes manifest dosage sensitivity in rich media[52], and therefore

DAmP may not in general provide a knockdown effect as strong as CRISPRi. This CRISPRi library has distinct advantages compared with currently available genome-wide methods—the gRNAs are designed based on S. cerevisiae specific rules, and more importantly, the repression is inducible, enabling control over the scale, context, and timing of gene perturbations. The ability to quantitatively probe the role of essential genes is also a major advantage of inducible CRISPRi over both uninducible CRISPRi and other systematic approaches. In fact, our outgrowth results demonstrate that even under conditions of minimal competition (colony growth) the gRNAs targeting haploinsufficient and dosage-sensitive genes are depleted with respect to both randomly shuffled gRNAs and gRNAs targeting non-dosage-sensitive genes. After 24 h of outgrowth with limited competition, the frequency of many of the gRNAs targeting dosage-sensitive genes had fallen by more than fourfold in the induced media. This means that after five library passages, the frequency of these gRNAs would fall to less than one thousandth of their original levels. In addition, competitive exponential growth in liquid media would exacerbate the fitness effects further. Therefore, without inducibility, it would be only a matter of days for the gRNAs targeting dosage-sensitive genes to drop-out of the assay. In addition, our use of 3D semisolid agarose to generate and interrogate large diverse libraries provides a more efficient approach over traditional 2D plating protocols while, at the same time, minimizing competitive biases that confound liquid outgrowth. Furthermore, the ability to easily transform the library into any genetic background of interest will enable rapid, parallel mapping of genetic interactions for any allele of interest[29].

Following our deposit of an earlier version of this manuscript on *biorXiv*, another group deposited a pre-print describing a similar inducible genome-scale CRISPRi library in yeast[53]. The library introduced by McGlincy et al.[53] and the presented library here share many similarities, including the general guide RNA design and the expression vector. But the library presented here provides distinct advantages—this library is more compact and less expensive to propagate and analyze. The library introduced here can be sequenced by performing 75 cycles of sequencing rather than 150, thereby reducing sequencing cost by ~50%. In addition, the use of 3D Gel instead of bioreactor makes our approach more broadly accessible. Furthermore, by averaging the strongest three spacers we are able to identify the affected genes in an unbiased fashion, capturing dosage sensitivity in a higher percentage of essential genes.

The presented CRISPRi library will provide a powerful and versatile tool for genetic interrogation of yeast biology, and we anticipate many applications across basic biology and biotechnology. For example, this library can be transformed into any mutant background and used to systematically study epistatic

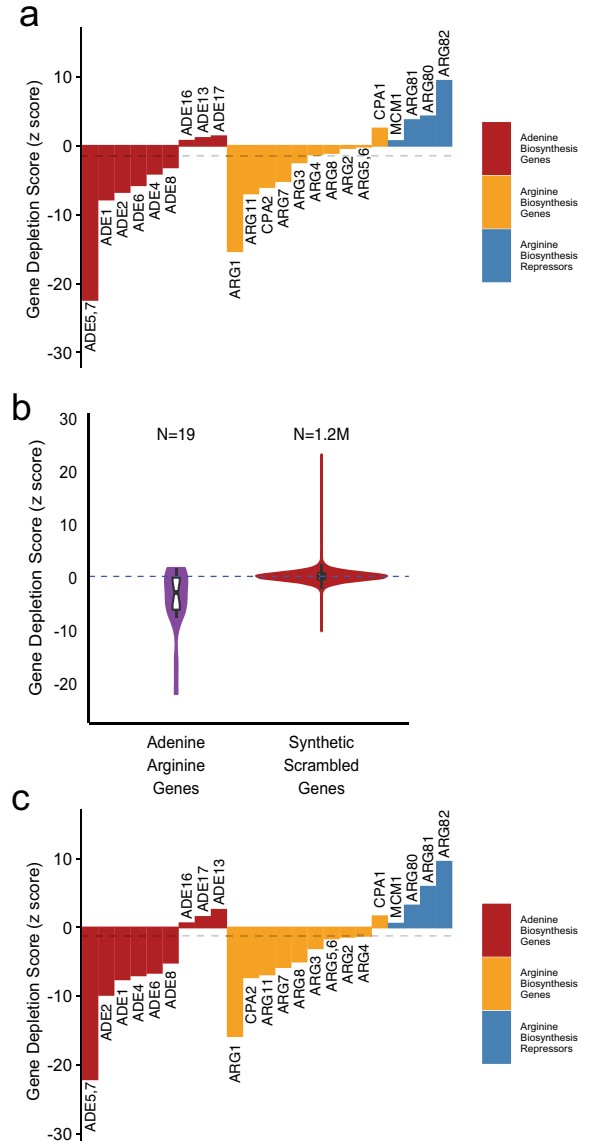

**Fig. 5 Identification of adenine and arginine biosynthetic genes. a** Bar plots showing the depletion scores for the known annotated genes involved in adenine and arginine biosynthesis pathways in addition to arginine regulatory genes. Dashed gray line marks the threshold for FDR = 10%. **b** Violin plots depicting the gene depletion score distribution for the adenine arginine deprivation experiment (SC-His-Ade-Arg +ATc vs. SC-His +ATc), shown for the known adenine/arginine biosynthetic genes, and synthetic scrambled genes, resulting from 200× random sampling of synthetic randomly shuffled gRNAs. **c** Bar plots showing the depletion scores for the annotated genes involved in adenine and arginine biosynthesis pathways in addition to arginine regulatory genes, restricted to gRNAs with PAM located within 150 bp of TSS. Dashed gray line marks the threshold for FDR = 10%.

interactions between the mutation(s) and all other gene perturbations. Our focused study of arginine/adenine deprivation, in fact, demonstrates attractive sensitivity/specificity characteristics for probing the genetic basis of arbitrary phenotypes and biological processes.

## Materials and methods
**Strains, plasmids, and media.** PRS416-MXI1 was ordered from AddGene (Cambridge, MA, USA). In order to create amPL43, the PRS416-MXI1 backbone was PCR amplified. His3 was amplified from pSH62[54]. The primers are listed in

Supplementary Data 7. Gibson assembly (NEB, Ipswich, MA, USA) was used to create amPL43. The plasmid's sequence was verified using Sanger sequencing. gRNA oligo library was purchased from Custom Array (92918-format chip, Bothell, WA, USA). The library spacers were amplified and extended using 5′-Library-Extender and 3′-Library-Extender primers as described by Smith et al.[22]. To generate the library, amPL43 was maxiprepped (Qiagen, Hilden, Germany) and cut with NotI. Gibson Assembly was used to clone the oligos in the NotI site of amPL43. The library was transformed into DH5α *E. coli* (NEB C2987H). The transformed bacteria were grown in 1L of semisolid LB plus 100 µg/ml ampicillin (0.35% Seaprep Agarose, Lonza) to minimize competition between the colonies. Use of semisolid media would minimize the competition and ensure that the growth of the colonies would be independent of each other and avoid any potential jackpotting effects. We sequenced 32 individual colonies to assess the quality of our cloning method. We did not detect any empty vector or repeat sequences. To pool the library, the semisolid media was stirred for 10 min using a magnetic stirrer. 50 ml of the library was collected by centrifugation. The pooled library was mini-prepped (Promega, Madison, WI, USA) according to manufacturer's instructions. Two independent library replicates were generated.

Competent BY4741 cells were produced using the standard LiAc/PEG method[34], with minor modifications: 5 mL of SC + glucose media was inoculated with a fresh colony of BY4741 and grown overnight in 30 °C shaker. 1 ml of overnight culture (OD$_{660}$ ~ 6) was added to 25 mL of SC + glucose and shaken in 30 °C shaker for 4 h (OD$_{660}$ ~ 0.7). Cells were pelleted by centrifugation at 3000 g for 5 min and washed twice in 25 mL sterile water. Next, cells were resuspended in 1 mL sterile water and transferred to 1.5 mL Eppendorf tube and centrifuged for 1 min at 4000 × g and then resuspended in sterile water to a final total volume of 1 mL. In all, 200 µL of competent cells were aliquoted into an Eppendorf tube and centrifuged for 30 s at 4000 g and the supernatant was removed. Next, 240 µL of freshly made sterile 50% PEG 3500, 36 µL of sterile 1 M lithium acetate, 50 µL of Salmon sperm DNA (Thermofisher, Waltham, MA, USA), and 2 µg of plasmid DNA was added to the cells and the volume was adjusted to 350 µL using sterile water. Then the mixture was vortexed and incubated at 42 °C for 20 min, vortexed, and again incubated at 42 °C for another 20 min. Transformed cells were centrifuged and supernatant was removed, and 1 mL of SC + glucose was added to it. The cells were moved to a round bottom falcon tube and shaken at 30 °C for 1 h.

We transformed three yeast replicates from the first bacterial library and two replicates from the second bacterial library. The transformed library was grown in semisolid SC-His+glu media (0.35% Seaprep Agarose) for 48 h, pooled, and resuspended in SC-His+glu media plus 20% glycerol and frozen for future use. To use the frozen library in a future experiment, we need to estimate the number of colony-forming units (CFU) per microliter in the frozen stock. To this end, we add 25 µL of the thawed library to 1 ml of SC-His+glu media and shake for 2 h at 30 °C. We, then, plate appropriate dilutions of the outgrowth to estimate the CFU of the frozen media. To use the frozen stock, we add appropriate amount of thawed stock to 10 ml of media and shake for 2 h at 30 °C before adding it to the semisolid media.

To extract the plasmids, 50 OD$_{660}$ of cells were pooled and resuspended in 2 ml SE buffer (0.9 M sorbitol, 0.1 M EDTA pH 8.0), 200 µl Zymolyase 100 T (2 mg/ml) and 2 µl β-mercaptoethanol and incubated at 37 °C for 1 h, followed by standard mini prep extraction per manufacturer's instructions (Promega). The pooled plasmid library was suspended in 50 µL of sterile water. For this study, we inoculated 250 ml of the following three semisolid media with a number of cells equivalent to 1 OD660 unit of the library: A) SC-His with ATc, B) SC-His-Arg-Ade with ATc C) SC-His-Arg-Ade without ATc.

To prepare the semisolid media, 0.35% of sea prep agarose and media mix was autoclaved, with the magnetic stir bar left in. The media was cooled down at room temperature. In our experience, 1L of media can hold up to ~1–5 million colonies. Therefore, the efficiency of transformation has to be determined. Appropriate amounts of transformed cells are moved to the semisolid media and are mixed using a magnetic stirrer for 10 min. The number of colonies can be estimated by growing 500 microliters of the media on a plate. The semisolid media is chilled in an ice-bath for 1 h to allow the media to gel. In our experience, the ice-bath level should be higher than the semisolid media. The media is transferred carefully to 30 °C or 37 °C incubator not to disrupt the gel. The individual colonies should be visible the next day for bacterial culture and in 2 days for yeast transformation.

Haploinsufficient genes were derived from Deutschbauer et al.[2]. Nonessential and essential gene lists were derived from Giaever et al.[3].

**qPCRS.** To measure the fold change in the target gene in the absence of induction, strains were typically grown in SC-HIS overnight, diluted to an OD660 of 0.07 without 250 ng/mL ATc. As control, we used a BY4741 strain without the sgRNA construct. We selected a mix of three different gRNAs per target gene. The sequences for the selected gRNAs are reported in Supplementary Data 7. The cells were grown at 30 °C and a sample was taken from each tube at 24 h. RNA was extracted using Norgen Biotek Total RNA Purification Kit (Norgen Biotek, ON, Canada) and cDNA was made using Maxima H First Strand cDNA Synthesis Kit, with dsDNase (Thermofisher). ERG25, ERG11, and Sec14 primers and their corresponding spacers were adapted from Smith et al.[22]. qRT-PCR was performed using SYBR® Green PCR Master Mix (Life Technologies, Carlsbad, CA, USA) and

the Quantstudio 5. ΔΔCt of the target genes' in induced versus uninduced states as compared with ACT1 level are reported. One of the replicates for Sec14 sample was excluded for being an outlier (not within mean ± 2× standard deviation).

To measure the fold change in the target gene as the result of induction, strains were typically grown in SC-HIS overnight, diluted to an OD660 of 0.07 with/without 250 ng/mL ATc. We selected a mix of three different gRNAs per target gene. The sequences for the selected gRNAs are reported in Supplementary Data 7. The cells were grown at 30 °C and a sample was taken from each tube at 1, 4, 7, and 24 h. RNA extraction, cDNA synthesis and qRT-PCR were performed as above.

**Design of the gRNA Library**. The gRNA sequences and their relative location to TSS, as well as the nucleosome occupancy scores, were adapted from Smith et al.[22]. When possible, the gRNAs that were located between 0 and 200 bp of the TSS, were sorted based on their nucleosome occupancy score and the top six gRNAs were chosen. In rare cases, when six gRNAs were not available for any given gene, we searched for the gRNAs further from TSS. When more than six acceptable gRNAs were available, up to 12 gRNAs were included.

**Next-generation sequencing and analysis**. The extracted plasmids were amplified using "lib-seq" primers in the Supplementary Data 7 for 10 cycles using Phusion PCR kit (NEB). Each 50 μL PCR reaction consisted of 6 μL of plasmid library, 10 μL of 5× Phusion HF buffer, 0.5 μL of Phusion DNA Polymerase, 10 μM Forward Primer, 10 μM Reverse Primer, and 10 mM dNTPs. Three replicate reactions per sample were amplified. The PCR conditions were: 98 °C for 1 min for initial denaturing, then 98 °C for 18 s, 66 °C for 18 s, and 72 °C for 30 s for 10 cycles, then 72 °C for 10 min. The products were purified using AMPure XP beads (1.4:1 bead to DNA ratio, Beckman-Coulter, Brea, CA, USA). The Illumina adapters and the indices were added by a second PCR using Q5 polymerase (NEB). Each 50 μL PCR consisted of 2–10 μL of purified PCR from the first PCR, 10 μL of 5× Q5 Reaction Buffer, 10 μL 5× Q5 High GC Enhancer, 0.5 μL of Q5 High-Fidelity DNA Polymerase, 10 μM Forward Primer, 10 μM reverse primer, 10 mM dNTPs. The PCRs were done in two steps. First step conditions were: 98 °C for 1 min for initial denaturing, then 98 °C for 18 s, 62 °C for 18 s, and 72 °C for 30 s for 4 cycles. Second stage of the PCR was 98 °C for 18 s, 66 °C for 18 s, and 72 °C for 30 s for variable cycles and finally 72 °C for 10 min. The number of cycles should be determined using a parallel qPCR to ensure that the sample does not saturate. The samples were purified using AMPure XP beads (1.3:1 bead to DNA ratio). The samples were run on an Illumina Hi-Seq 4000 for 75 cycles paired-end with a 58–17 breakdown for read 1 and read 2. We used Cutadapt and Bowtie2 to trim the sequences and map them to the targeting sequences with a maximum of one mismatch[55,56]. We calculated the log frequency of the reads as:

$$log(f) = log_2 \left( \frac{\text{No. of read} + 1}{\text{Total number of reads for that sample}} \right) \quad (1)$$

To calculate the depletion scores, we filtered the gRNAs to have a minimum read frequency of $10^{-5}$ in either the treatment or the control group, i.e., they needed to have a minimum number of reads in the induced or the uninduced sample. The frequency of reads, the number of gRNAs with a minimum of one read, and the number of gRNAs passing the threshold are depicted in Supplementary Fig. 2. When a gRNA is common to two genes, we assigned its effect to both genes. The distribution of gRNAs per gene in each sample is shown in Supplementary Fig. 4. The gRNA depletion/enrichment score between sample A and B was calculated using:

$$gRNA\ depletion\ score = log(f_A) - log(f_B) \quad (2)$$

To assign a gene depletion or enrichment score, we sorted the gene log-fold change values of the gRNAs targeting a gene. Then, we averaged the highest top three values and also bottom three values. We compared the absolute numbers for the top three and the bottom three gRNAs targeting any given gene and we assigned the largest absolute value as the depletion/enrichment score for that gene. If there were less than six but more than three gRNAs present for a gene, for example, $g_{1-4}$ sorted based on their values, we compared the average for $g_{1-3}$ and $g_{2-4}$. For genes with three or less gRNAs, we assigned the average of the gRNAs as the gene depletion score. Previous studies[26] used a metric of average phenotype for the top three most effective gRNAs for each gene. The method presented here has the benefit that it would account for the possibility that repression of some genes could, in fact, be beneficial for growth.

To create the synthetic scrambled genes, we utilized the gRNA depletion scores of the synthetic randomly shuffled gRNAs. We simulated a pool of synthetic scrambled genes by sampling the depletion scores of the synthetic randomly shuffled gRNAs, whereas maintaining the distribution of gRNA per gene in the sequenced CRISPRi library. Synthetic scrambled genes were populated by considering each yeast ORF and replacing its corresponding gRNAs depletion scores from the pool of synthetic randomly shuffled gRNAs. This process was repeated 200 times to generate a distribution of synthetic scrambled genes. The gene depletion scores were converted to z scores based on the distribution of synthetic scrambled genes in that replicate. The gene scores (z scores) for the replicates were then averaged using Stouffer's method.

We used the depletion z score to classify each gene. For a given z score threshold, we considered genes with depletion scores below that threshold to be hits (e.g., dosage sensitive). We assessed true positives based on the genes known to be in a given category (e.g., haploinsufficient or essential genes) and false positives based on the pool of synthetic scrambled genes. True positive rate and false positive rates were calculated for a range of z score thresholds and the Area Under the ROC Curve was assessed. Given the uneven numbers of positives and negatives, we calculated the FDR for each decision value (threshold) using:

$$FDR = \frac{\text{True Positives (TP)}}{\text{True Positives + False Positives (FP)}} \quad (3)$$

where we oversampled the positive genes.

Secondary structure free energy was calculated using RNAfold application from ViennaRNA package 2.4.14 using default parameters[57]. The full length sgRNA was assigned from the approximate TSS ("gtccctatcagtgatagaga") to the end of sgRNA scaffold at "tcggtgctttttctcgag".

**IPAGE**. We ran iPAGE[37] in continuous mode on the average depletions/enrichment scores of guide RNAs calculated at the gene level as described above. iPAGE discovers gene categories that are significantly informative (p value < 0.05, random shuffling) of the average scores. Scores were sorted in 7 bins and only biological processes categories were considered.

**Random forest classifier**. As a proxy for sgRNA activity, we collected the $Log_2$(gRNA Depletion Score) of 3507 sgRNAs corresponding to markedly depleted essential and haploinsufficient genes from the outgrowth experiment (SC-HIS +ATc vs SC-HIS -ATc). We then retained two categories for the most efficient (lowest 20% $Log_2$(gRNA Depletion Score)) and least efficient crRNAs (top 20% $Log_2$(gRNA Depletion Score)). As features for prediction, we considered the distance of the PAM of every sgRNA from the TSS (absolute distance and the sign of the distance), the nucleosome occupancy score[22], the free energy of the sgRNA transcript (calculated using RNAfold), whether the PAM is on the same or opposite strand as the gene, the GC content of the 20 nucleotides of the sgRNA, and the size of the longest homo-polymer for each nucleic acid (A, C, G, T). Individual bases in the 20 nucleotides of the sgRNA as well as 10 nucleotides upstream and downstream of the target sequence on the genome were considered as features. The nucleotide space was represented with one-hot encoding allowing for independent weights for each possible base[58], and both mono-nucleotides and di-nucleotides were considered at every position.

To predict whether a given sgRNA has efficient inhibitory activity or not, we trained a Random Forest model to classify gRNAs from the most and least efficient sgRNA categories using the Random Forest[59] scikit-learn python package. We estimated the overall performance of the classifier under threefold cross-validation to prevent overfitting the model. For every training set, the features space was reduced to 130 features using univariate analysis with a chi2 metric (SelectBestK python routine) and the Random Forest model was trained using these features. ROC and Precision–recall curves were calculated by varying the predict class probability and using the three cross-validation models assessed on their respective test sets. The process was repeated 10 times for different cross-validation partitions to estimate errors for AUC and AUPRC. We assessed the contribution, $C_f$, of each feature by estimating the average accuracy of the model on the test set when all values of a given feature are permuted across sgRNAs. This measure was converted to a percentage contribution, $P_f$, by comparison to the average accuracy of the unpermuted dataset ($C_{base}$) and the average accuracy of the fully permuted dataset for all features ($C_{min}$):

$$P_f = 100 \times \frac{C_{base} - C_f}{C_{base} - C_{min}} \quad (4)$$

which reflects the contribution of each feature to the model as a percentage of the total accuracy of the model. To calculate the contribution of each individual base position to the model, we permuted the nucleotides in that particular base and recalculated the one-hot encoding for the mono- and di- nucleotides of the test sets. All permutations were repeated 30 times and averages were calculated for each model. The process was repeated for each cross-validation model and each cross-validation partition.

**Reporting summary**. Further information on research design is available in the Nature Research Reporting Summary linked to this article.

## Data availability
All data generated or analyzed during the course of this study are included in the Supplementary Data associated with this manuscript (See Supplementary Data 1–7 for details). The raw sequencing data that support the findings are available at GEO accession: GSE159409. The two libraries are biological replicates. CRISPRi library replicate 1 was deposited to Addgene (Addgene ID: 161829). The library replicate 2 will be available upon request from the corresponding author. AmPl43 was also deposited to Addgene (Addgene ID: 161830).

## Code availability

The custom developed code to analyze the sequencing data is available on our GitHub at https://github.com/tavalab/yeast-CRISPRi.

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

## Acknowledgements

We thank members of the Tavazoie laboratory for helpful discussions and feedback on the manuscript, especially Wenyan Jiang and Balaji Santhanam. A.M.R was supported by a Ruth Kirschstein NRSA Postdoctoral Fellowship Award (F32-GM125170) from NIGMS. S. T. was supported by grants from the NIH (R01-AI077562 and R01-HG009065). pRS416-dCas9-Mxi1 + TetR + pRPR1(TetO)-NotI-gRNA was a gift from Ronald Davis (Addgene plasmid # 73796; http://n2t.net/addgene:73796; RRID:Addgene_73796).

## Author contributions

A.M.R. conceived the study and designed the experiments with the help of P.O. and S.T. A.M.R. and P.O. performed the experiments with the help of M.Z. A.M.R. and P.O. analyzed the results. A.M.R., P.O., and S.T. wrote the manuscript. All authors read and approved the final manuscript.

## Competing interests

The authors declare no competing interests.

## Additional information

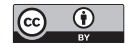

