## [Peer Review File · Communications Biology]

Reviewers' Comments:

Reviewer #1:

Remarks to the Author:

This manuscript reports development of inducible yeast CRISPRi library and applications of this for functional genomics study. Followings are very important and advantage points of this manuscript: 1) this is the first report of developing a genome wide inducible CRISPRi library for *S. cerevisiae*, 2) improved the gRNA design rules in yeast, 3) showed that this library had a very strong potential to perform versatile functional genomics study such as identification of genome wide dosage sensitive genes or essential genes and genes involved in specific biosynthesis pathways in yeast, 4) reported usefulness of 3D semisolid agarose culture. Specifically, as mentioned in the manuscript, introduction of inducible system in CRISPRi library is highly evaluated because this system can regulate the timing of gene inactivation. Quality of data in the manuscript is good and text and figures are well organized. I think this is a great work and this library will greatly contribute to and promote yeast genome and gene analyses. On the other hand, the point that should be addressed is there are just a few examples or detailed idea of future usage of this library in the manuscript. I think it is very important to show authors' detailed idea of how this library can be involved in and contribute to yeast bioscience in the future.

Reviewer #2:

Remarks to the Author:

Momen-Roknabadi and Oikonomou et al. have constructed an inducible CRISPRi library as a new platform for yeast genetics. Yeast of course is a historically dominant genetic model organism and so this approach may enable new discoveries. Problematically the paper doesn't seem to present any new biological findings and so the novelty entirely stems from the CRISPRi sgRNA library as a method and resource. The manuscript would benefit from a clear demonstration of why an inducible sgRNA library reveals biology missed by other approaches such as more traditional yeast genetic arrayed libraries of KO or KD (TS or DAMP alleles) as well as newer systems like the MAGIC CRISPRi manuscript that the authors cite.

The findings at hand are claimed to be novel in two ways:

One is that the new library presented in this manuscript is inducible. The authors should annotate at a gene or sgRNA level how an inducible sgRNA library enables them to do something new or detect some biology related to essential or dosage sensitive genes that was not previously possible using published libraries such as referenced in Lian et al (reference 25). One could imagine showing that the MAGIC library loses sensitivity pre-T0 reference point on essential or dosage sensitive gene phenotypes.

The second novelty relates to new design rules that the authors claim are somewhat yeast specific. To maximize the utility of their work the authors should either provide an in silico prediction of maximally active sgRNAs target all genes or even better design a V2 library and demonstrate experimentally that their predictions for highly active sgRNAs hold true for sgRNA not present in the current library. The yeast specific rules highlighted in this manuscript relate to positional rules for targeting CRISPRi relative to the TSS but ignore rules highlighted in human libraries such as sequence composition, nucleotides at each position, nucleotide dimers, nucleotide homopolymers, RNA folding and more. Are these other features not important in yeast? Would be good to have a more complete assessment of the features predictive of highly active sgRNAs that are highlighted in the Horlbeck et al 2016 elife manuscript.

Other points:

The value of this paper as a resource stems in large part from these reagents being freely

accessible and so the authors should commit to making the yeast and DNA libraries readily available through SGD (or some other repository) and Addgene.

Is there anything biologically novel about the results of the arginine or adenine screens?

In Figure 2D please comment on the difference in the distributions between the non-essential genes and the synthetic scrambled controls. Is the signal in the non-essential distribution due to neighbor genes that are dosage sensitive or essential? Or is this some other off target activity? Or do the sgRNA that deplete target genes that were previously unidentified as necessary for cell proliferation?

The statistical analysis is all appropriate and valid.

It is worth noting that another genome scale inducible CRISPRi sgRNA library is on biorxiv (<https://www.biorxiv.org/content/10.1101/2020.03.11.988105v1.full.pdf>).

Reviewer #3:

Remarks to the Author:

In this manuscript Momen-Roknabadi et al. describe a CRISPRi system for CRISPR-dCas9 based genetic repression in *S. cerevisiae* yeast, and further describe some novel methodology and principles of design for CRISPRi genetic repression in yeast. The authors construct an inducible, genome-wide repression library and as proof-of-concept, identify genes involved in arginine and adenine biosynthesis. This study presents interesting new methods and design principles for CRISPRi and is very well written. But more clarity should be provided to contextualize this work in the field and clarify how it differs from and improves upon previous research.

Specific points

1. Given the focus of this manuscript is on CRISPRi, I think the impact could be improved by having a more detailed introduction to CRISPRi in the introduction section. While the authors do describe CRISPRi generally (mostly in terms of technology), it would be worthwhile to include much more detail on how this technology has been exploited in yeast, other fungi, and/or mammalian systems, and the kinds of important applications/discoveries that have been made using this technique. Essentially, it would be nice to hear about why this technology is relevant and powerful, and what sort of exciting applications it has in diverse systems.
2. My main concern is that it is difficult to discern from the writing what is novel about this study and where it fits into the landscape of other CRISPRi in yeast studies. The authors emphasize the importance of the inducible tetO promoter, however this was in fact used in other studies (Smith et al). In the introduction the authors compare theirs to the MAGIC system - but this was a very different study that analyzed gain-of-function, reduction-of-function, and loss-of-function libraries. Is the difference that this manuscript is the first to use the inducible system on a genome-wide level? If so, this should be clarified and emphasized.
3. The 3D media selection seems to be a novel aspect of this method, and it would be appropriate to highlight and emphasize this part more, and perhaps include some additional experimental detail in the results or methods section. It may also be helpful to include an experimental design figure in supplemental that describes the workflow including this 3D media selection component.
4. For Figure 1b - it would be helpful to visualize this as each time point - or + Atc, instead of just showing fold difference. It would allow one to better see the 'leakiness' of the promoter, and how repression is affected even without the inducer.
5. A strength of this manuscript is the emphasis on validation of their system at different steps along the way, as well as good incorporation of controls, replicates, and statistics.

Minor points

1. The authors capitalize Adenine and Arginine throughout, and I do not believe they should be.
2. There is a section titled "Results and Discussion" and another "Discussion"
3. Some parts of the Results section include unnecessary experimental detail, which belongs in the materials and methods section (ie E. coli (NEB C2987H....); 0.35% seaprep agarose...)
4. Line 150 states that "we inoculated media with distinct gRNA targets", which doesn't make sense.

We want to thank the reviewers for their careful reading of the manuscript, their supportive comments and their helpful suggestions. We have addressed all reviewer comments with additional analysis, experiments and/or modifications to the revised manuscript. We have underlined the text with substantial changes in the manuscript. In our detailed responses below, we provide the reviewers' comments as *italicized text*, our responses in normal text, and specific modifications to the manuscript as underlined text.

Reviewer #1 (Remarks to the Author):

*This manuscript reports development of inducible yeast CRISPRi library and applications of this for functional genomics study. Followings are very important and advantage points of this manuscript: 1) this is the first report of developing a genome wide inducible CRISPRi library for *S. cerevisiae*, 2) improved the gRNA design rules in yeast, 3) showed that this library had a very strong potential to perform versatile functional genomics study such as identification of genome wide dosage sensitive genes or essential genes and genes involved in specific biosynthesis pathways in yeast, 4) reported usefulness of 3D semisolid agarose culture. Specifically, as mentioned in the manuscript, introduction of inducible system in CRISPRi library is highly evaluated because this system can regulate the timing of gene inactivation. Quality of data in the manuscript is good and text and figures are well organized. I think this is a great work and this library will greatly contribute to and promote yeast genome and gene analyses. On the other hand, the point that should be addressed is there are just a few examples or detailed idea of future usage of this library in the manuscript. I think it is very important to show authors' detailed idea of how this library can be involved in and contribute to yeast bioscience in the future.*

We thank the reviewer for their encouraging comments. We have now added more details on the advantages of this library and how it can be used to further contribute to our understanding yeast biology throughout the manuscript. Below are two specific examples:

Compared with the traditional genome engineering techniques, such as knock out collections, CRISPRi enables the systematic interrogation of all biological processes under different genetic backgrounds and environmental conditions. This technology has been applied in wide range of organisms from bacteria to human cell lines, to downregulate the expression of both essential and nonessential genes. This has enabled a diverse set of studies from characterizing the role of long non-coding RNAs, to identifying the contributing factors to drug resistance, and many other biological phenomena.

The presented CRISPRi library will provide a powerful and versatile tool for genetic interrogation of yeast biology, and we anticipate many applications across basic biology and biotechnology. For example, this library can be transformed into any mutant background and used to systematically study epistatic interactions between the mutation(s) and all other gene perturbations. Our focused study of arginine/adenine deprivation, in fact, demonstrates attractive sensitivity/specificity characteristics for probing the genetic basis of arbitrary phenotypes and biological processes.

Reviewer #2 (Remarks to the Author):

Momen-Roknabadi and Oikonomou et al. have constructed an inducible CRISPRi library as a new platform for yeast genetics. Yeast of course is a historically dominant genetic model organism and so this approach may enable new discoveries. Problematically the paper doesn't seem to present any new biological findings and so the novelty entirely stems from the CRISPRi sgRNA library as a method and resource. The manuscript would benefit from a clear demonstration of why an inducible

sgRNA library reveals biology missed by other approaches such as more traditional yeast genetic arrayed libraries of KO or KD (TS or DAMP alleles) as well as newer systems like the MAGIC CRISPRi manuscript that the authors cite.

We thank the reviewer for their constructive comments. The main novelty of this manuscript is in the approach itself and our efforts to make the method accessible to the scientific community. In addition, our paper, thanks to the reviewers, refines the rules governing the design of gRNAs for yeast. Finally, we have now added a section describing the advantages of this method over other approaches:

Genetic studies in *Saccharomyces cerevisiae* benefit from a wide array of techniques for studying loss-of-function phenotypes. Some of the most widely used methods to study loss of function in budding yeast are gene deletion/knock-out, temperature-sensitive mutants, and DAmP mutants. Temperature-sensitive mutations have also been widely used. However, temperature-sensitive mutations are difficult to construct in a systematic manner. Gene deletion libraries can be constructed in a systematic manner, although essential genes will not be covered, at least, in a haploid background. Finally, in DAmP library assays, only 17% of the viable alleles for essential gene manifest dosage-sensitivity in rich media (52), and therefore DAmP may not in general provide a knockdown effect as strong as CRISPRi. This CRISPRi library has distinct advantages compared to current available genome-wide methods: The gRNAs are specifically designed based on *S. cerevisiae* specific rules, and more importantly, the repression is inducible, enabling control over the scale, context, and timing of gene perturbations. The ability to quantitatively probe the role of essential genes is also a major advantage of inducible CRISPRi over both uninducible CRISPRi and other systematic approaches. In fact, our outgrowth results demonstrate that even under conditions of minimal competition (colony growth) the gRNAs targeting haploinsufficient and dosage-sensitive genes are depleted with respect to both randomly shuffled gRNAs and gRNAs targeting non-dosage-sensitive genes. After 24 hours of outgrowth with limited competition, the frequency of many of the gRNAs targeting dosage-sensitive genes had fallen by more than four-fold in the induced media. This means that after five library passages the frequency of these gRNAs would fall to less than one thousandth of their original levels. In addition, competitive exponential growth in liquid media would exacerbate the fitness effects further. Therefore, without inducibility, it would be only a matter of days for the gRNAs targeting dosage-sensitive genes to drop out of the assay. In addition, our use of 3D semisolid agarose to generate and interrogate large diverse libraries provides a more efficient approach over traditional 2D plating protocols while, at the same time, minimizing competitive biases that confound liquid outgrowth. Furthermore, the ability to easily transform the library into any genetic background of interest will enable rapid, parallel mapping of genetic interactions for any allele of interest (29).

The findings at hand are claimed to be novel in two ways:

One is that the new library presented in this manuscript is inducible. The authors should annotate at a gene or sgRNA level how an inducible sgRNA library enables them to do something new or detect some biology related to essential or dosage sensitive genes that was not previously possible using published libraries such as referenced in Lian et al (reference 25). One could imagine showing that the MAGIC library loses sensitivity pre-T0 reference point on essential or dosage sensitive gene phenotypes.

We thank the reviewer for giving us the opportunity to distinguish our library from Lian *et al.*¹. Our library has the following advantages:

- 1) The first difference between our approach and that of MAGIC, is that MAGIC is not based on the design rules specific for yeast, making the library design less than ideal.

¹ Jiazhang Lian et al., "Multi-Functional Genome-Wide CRISPR System for High Throughput Genotype-Phenotype Mapping," *Nature Communications* 10, no. 1 (December 19, 2019): 5794–10, doi:10.1038/s41467-019-13621-4.

- 2) Unlike the library presented here, MAGIC is not inducible. Therefore, the library could accumulate suppressor mutations. More importantly, the MAGIC library could lose its sensitivity pre-T0, as the cells harboring the gRNAs targeting dosage-sensitive genes will be less competitive. Unfortunately, it is not possible for us to calculate the loss of sensitivity directly based on the data available from Lian *et al.* To calculate the loss, one must be able to compare with the library composition before and after the two-day outgrowth. Unfortunately, the composition of the library before the outgrowth is not readily available. However, we are able to estimate the loss of gRNA diversity using our library (see below).
- 3) In addition, our library is comprised of more gRNAs (more than 51,000 designed gRNAs as opposed to 37,870 gRNAs in MAGIC).
- 4) Finally, our library can be sequenced using 75 cycles as opposed to 160 cycles in MAGIC, making the screen less expensive to sequence.

In fact, our outgrowth results demonstrate that even under conditions of minimal competition (colony growth) the gRNAs targeting haploinsufficient and dosage-sensitive genes are depleted with respect to both randomly shuffled gRNAs and gRNAs targeting non-dosage-sensitive genes. After 24 hours of outgrowth with limited competition, the frequency of many of the gRNAs targeting dosage-sensitive genes had fallen by more than four-fold in the induced media. This means that after five library passages the frequency of these gRNAs would fall to less than one thousandth of their original levels.

The second novelty relates to new design rules that the authors claim are somewhat yeast specific. To maximize the utility of their work the authors should either provide an in silico prediction of maximally active sgRNAs target all genes or even better design a V2 library and demonstrate experimentally that their predictions for highly active sgRNAs hold true for sgRNA not present in the current library. The yeast specific rules highlighted in this manuscript relate to positional rules for targeting CRISPRi relative to the TSS but ignore rules highlighted in human libraries such as sequence composition, nucleotides at each position, nucleotide dimers, nucleotide homopolymers, RNA folding and more. Are these other features not important in yeast? Would be good to have a more complete assessment of the features predictive of highly active sgRNAs that are highlighted in the Horlbeck et al 2016 elife manuscript.

We thank the reviewer for their insightful comment and suggestions. We have now developed a random forest classifier to predict the efficacy of the sgRNAs. Random forests reduce the overfitting by randomly selecting a subset of features. In addition, these classifiers are robust to outliers because they bin them. In line with the findings by Horlbeck et al² we find that the distance to the TSS is the most important factor. However, our model further demonstrates that the rules governing the activity of the sgRNAs in yeast are unique to *S.cerevisiae*:

Overall, the distance of PAM to the TSS had the greatest percentage contribution (~54% - 65%, Wilcoxon signed-rank test p -value $<10^{-10}$), while overall sequence features represented the second largest effect on the model (~34% - 49%, Wilcoxon signed-rank test p -value $<10^{-10}$). Nucleosome occupancy contributed to a lesser extent (~2% - 8%, Wilcoxon signed-rank test p -value $<10^{-3}$), while additional individual parameters including gRNA secondary structure, strandedness and GC content were not deemed as significant by the model (Wilcoxon signed-rank test p -value >0.1). We next investigated the contribution of each nucleotide at each location between 10 bp upstream and downstream of the target spacer (Figure 4e-f). This analysis demonstrated that the bases located at 1, 2, 4, 9, 10, 11, 12 and 20 base pairs upstream of PAM contribute significantly to the efficacy of

² M A Horlbeck et al., "Compact and Highly Active Next-Generation Libraries for CRISPR-Mediated Gene Repression and Activation," *Elifesciences.org*, n.d., doi:10.7554/eLife.19760.001.

gRNA (hypergeometric test, p -value <0.01). Out of these positions, gRNAs with an “A” at position -9 are predicted to have the highest contribution to an effective gRNA. In addition, the gRNAs without a poly-T stretch are more effective.

Other points:

The value of this paper as a resource stems in large part from these reagents being freely accessible and so the authors should commit to making the yeast and DNA libraries readily available through SGD (or some other repository) and Addgene.

We wholeheartedly agree with the reviewer and we have committed to deposit the more diverse library to Addgene. Since the libraries are very similar and the second library replicate is less diverse, it will be provided upon request.

Is there anything biologically novel about the results of the arginine or adenine screens?

The screen presented here detected most of the genes involved in adenine and arginine biosynthesis pathway in a single step. The presented screen also identified other genes with significantly depleted gRNAs. To analyze these genes in a systematic fashion, we used iPAGE to identify the pathways that were significantly affected. By carefully studying the affected pathways, we concluded that the affected biological processes, such as protein sorting pathways, are known to be impacted by disruptions in adenine or arginine biosynthesis. While there is no new biological insight gained from the adenine and arginine screen, this experiment serves two important purposes:

- 1) One of the advantages of this library is the ability to study its effect under an arbitrary condition, to discover the role of affected genes. This experiment shows that it is possible to use a simple drop-out strategy to study yeast biology using CRISPRi
- 2) Using this approach, we were able to use the drop-out media to successfully interrogate not only one but two different biological processes. This is important because it shows that this approach could easily be adapted to study the interactions between two seemingly unrelated biological processes.

In Figure 2D please comment on the difference in the distributions between the non-essential genes and the synthetic scrambled controls. Is the signal in the non-essential distribution due to neighbor genes that are dosage sensitive or essential? Or is this some other off target activity? Or do the sgRNA that deplete target genes that were previously unidentified as necessary for cell proliferation?

In yeast, a gene is classified as a non-essential gene under a certain condition if its null mutant can survive in that specific condition, with no regards for dosage-sensitivity of the gene. It has been shown previously that a small number (86 genes) of non-essential genes are haploinsufficient³. Here, we detected that around 35% of non-essential genes are, in fact, dosage-sensitive. However, this is in line with the previous findings that many of the non-essential genes are dosage-sensitive^{4,5}.

Surprisingly, our analysis showed that 35% of non-essential genes show dosage-sensitivity. However, 76% of the non-essential genes detected here as dosage-sensitive have been previously shown to decrease fitness when mutated (Supplementary table 4). Therefore, our results are in-line

³ Adam M Deutschbauer et al., “Mechanisms of Haploinsufficiency Revealed by Genome-Wide Profiling in Yeast,” *Genetics* 169, no. 4 (April 2005): 1915–25, doi:10.1534/genetics.104.036871.

⁴ Wenfeng Qian et al., “The Genomic Landscape and Evolutionary Resolution of Antagonistic Pleiotropy in Yeast,” *Cell Reports* 2, no. 5 (November 29, 2012): 1399–1410, doi:10.1016/j.celrep.2012.09.017.

⁵ David K Breslow et al., “A Comprehensive Strategy Enabling High-Resolution Functional Analysis of the Yeast Genome,” *Nature Methods* 5, no. 8 (August 2008): 711–18, doi:10.1038/nmeth.1234.

with a low false discovery rate, bolstering the utility of this library for systematic genetic analysis of phenotypes.

The statistical analysis is all appropriate and valid. It is worth noting that another genome scale inducible CRISPRi sgRNA library is on biorxiv:

<https://www.biorxiv.org/content/10.1101/2020.03.11.988105v1.full.pdf>).

We thank the reviewer for bringing this paper to our attention. Our library was deposited on *bioRxiv* before the cited paper here. We have now provided a comparison between our method and the above paper:

Following our deposit of an earlier version of this manuscript on *bioRxiv*, another group deposited a pre-print describing a similar inducible genome-scale CRISPRi library in yeast (53). The library introduced by McGlincy *et al* and the presented library here share many similarities, including the general guide RNA design and the expression vector. But the library presented here provides distinct advantages: this library is more compact and less expensive to propagate and analyze. The library introduced here can be sequenced by performing 75 cycles of sequencing rather than 150, thereby reducing sequencing cost by ~50%. In addition, the use of 3D Gel instead of bioreactor makes our approach more broadly accessible. Furthermore, by averaging the strongest three spacers we are able to identify significantly affected genes in an unbiased fashion, capturing dosage-sensitivity in a higher percentage of essential genes.

Reviewer #3 (Remarks to the Author):

*In this manuscript Momen-Roknabadi et al. describe a CRISPRi system for CRISPR-dCas9 based genetic repression in *S. cerevisiae* yeast, and further describe some novel methodology and principles of design for CRISPRi genetic repression in yeast. The authors construct an inducible, genome-wide repression library and as proof-of-concept, identify genes involved in arginine and adenine biosynthesis. This study presents interesting new methods and design principles for CRISPRi and is very well written. But more clarity should be provided to contextualize this work in the field and clarify how it differs from and improves upon previous research.*

Specific points

1. Given the focus of this manuscript is on CRISPRi, I think the impact could be improved by having a more detailed introduction to CRISPRi in the introduction section. While the authors do describe CRISPRi generally (mostly in terms of technology), it would be worthwhile to include much more detail on how this technology has been exploited in yeast, other fungi, and/or mammalian systems, and the kinds of important applications/discoveries that have been made using this technique. Essentially, it would be nice to hear about why this technology is relevant and powerful, and what sort of exciting applications it has in diverse systems.

We thank the reviewer for this helpful suggestion. We have now added more details about CRISPRi's applications. We now cite relevant studies in different organisms and also provide more details on the advantages of CRISPRi compared with other methods in the introduction and in the discussion.

Compared with the traditional genome engineering techniques, such as knock out collections, CRISPRi enables the systematic interrogation of all biological processes under different genetic backgrounds and environmental conditions. This technology has been applied in wide range of organisms from bacteria to human cell lines, to downregulate the expression of both essential and nonessential genes. This has enabled a diverse set of studies from characterizing the role of long

non-coding RNAs, to identifying the contributing factors to drug resistance, and many other biological phenomena.

The presented CRISPRi library will provide a powerful and versatile tool for genetic interrogation of yeast biology, and we anticipate many applications across basic biology and biotechnology. For example, this library can be transformed into any mutant background and used to systematically study epistatic interactions between the mutation(s) and all other gene perturbations.

2. My main concern is that it is difficult to discern from the writing what is novel about this study and where it fits into the landscape of other CRISPRi in yeast studies. The authors emphasize the importance of the inducible tetO promoter, however this was in fact used in other studies (Smith et al). In the introduction the authors compare theirs to the MAGIC system - but this was a very different study that analyzed gain-of-function, reduction-of-function, and loss-of-function libraries. Is the difference that this manuscript is the first to use the inducible system on a genome-wide level? If so, this should be clarified and emphasized.

We apologize about this ambiguity. We have adjusted the language to make this clear.

Although Smith et al, introduced a limited diversity inducible CRISPRi system, a genome-wide inducible CRISPRi library is lacking. Here, we introduce an inducible genome-scale library, dedicated for CRISPRi in *S. cerevisiae*, and designed based on the rules described previously.

3. The 3D media selection seems to be a novel aspect of this method, and it would be appropriate to highlight and emphasize this part more, and perhaps include some additional experimental detail in the results or methods section. It may also be helpful to include an experimental design figure in supplemental that describes the workflow including this 3D media selection component.

We are grateful for this helpful comment. We have now added a detailed protocol on the use of the 3D media and also included a schematic figure (supplementary figure 1)

To prepare the semisolid media, 0.35% of sea prep agarose and media mix was autoclaved, with the magnetic stir bar left in. The media was cooled down at room temperature. In our experience, 1L of media can hold up to roughly 1-5 million colonies. Appropriate amounts of transformed cells are moved to the semisolid media and is mixed using a magnetic stirrer for 10 minutes. The number of colonies can be estimated by growing 500 microliters of the media on a plate. The semisolid media is chilled in an ice-bath for 1 hour to allow the media to gel. The media is transferred carefully to 30°C or 37°C incubator not to disrupt the gel. The individual colonies should be visible the next day for bacterial culture and in two days for yeast transformation.

4. For Figure 1b - it would be helpful to visualize this as each time point - or + Atc, instead of just showing fold difference. It would allow one to better see the 'leakiness' of the promoter, and how repression is affected even without the inducer.

We completely agree with the reviewer. The experiment designed for the original figure 1b compared the effect of induced and uninduced samples. Therefore, it is not possible to use that data to measure the "leakiness" of the promoter. Previous studies had shown that the background expression of this promoter is low⁶. In addition, we have now added a new panel exploring the leakiness of the

⁶ Fahim Farzadfard, Samuel D Perli, and Timothy K Lu, "Tunable and Multifunctional Eukaryotic Transcription Factors Based on CRISPR/Cas.," *ACS Synthetic Biology* 2, no. 10 (October 18, 2013): 604–13, doi:10.1021/sb400081r.

promoter in our system (figure 1b). Consistent with previous studies, the uninduced activity of the promoter in our system is low.

5. A strength of this manuscript is the emphasis on validation of their system at different steps along the way, as well as good incorporation of controls, replicates, and statistics.

We thank the reviewer for their encouraging remarks.

Minor points

1. The authors capitalize Adenine and Arginine throughout, and I do not believe they should be.

We thank the reviewer for their criticism. We have now fixed this issue.

2. There is a section titled “Results and Discussion” and another “Discussion”

We thank the reviewer for pointing this out. We have now fixed this issue.

3. Some parts of the Results section include unnecessary experimental detail, which belongs in the materials and methods section (ie E. coli (NEB C2987H...); 0.35% seaprep agarose...)

We thank the reviewer for this helpful comment. We have now fixed this issue.

4. Line 150 states that “we inoculated media with distinct gRNA targets”, which doesn’t make sense.

We regret this sentence which is confusing. That sentence is now clarified.

To this end, we inoculated semisolid media with the equivalent of 1 OD660 of the library (with an average of ~450 copies of each library member) with and without ATc induction. The use of semisolid media reduces direct competition between strains, helping to maintain a more uniform representation of gRNAs.

Reviewers' Comments:

Reviewer #2:

Remarks to the Author:

the authors have addressed all my concerns.

Reviewer #3:

Remarks to the Author:

The authors have nicely addressed all of the points raised. I have no further comments on this manuscript, which will be a very nice addition to the field and a very useful tool.